# The T cell receptor repertoire of tumor infiltrating T cells is predictive and prognostic for cancer survival

Sara Valpione[1,2], Piyushkumar A. Mundra[1], Elena Galvani[1], Luca G. Campana [3], Paul Lorigan [2], Francesco De Rosa [4], Avinash Gupta[2], John Weightman[5], Sarah Mills[6], Nathalie Dhomen[1] & Richard Marais [1✉]

Tumor infiltration by T cells is paramount for effective anti-cancer immune responses. We hypothesized that the T cell receptor (TCR) repertoire of tumor infiltrating T lymphocytes could therefore be indicative of the functional state of these cells and determine disease course at different stages in cancer progression. Here we show that the diversity of the TCR of tumor infiltrating T cell at baseline is prognostic in various cancers, whereas the TCR clonality of T cell infiltrating metastatic melanoma pre-treatment is predictive for activity and efficacy of PD1 blockade immunotherapy.

[1] Molecular Oncology Group, Cancer Research UK Manchester Institute, The University of Manchester, Alderley Park, Macclesfield, Cheshire, UK. [2] Medical Oncology, The Christie NHS Foundation Trust, Manchester, UK. [3] Department of Surgery, The Christie NHS Foundation Trust, previously Department of Surgical Oncological and Gastroenterological Sciences DISCOG (University of Padova), The Christie NHS Foundation Trust, Manchester, UK. [4] Immunotherapy - Cell Therapy and Biobank, Istituto Scientifico Romagnolo per lo Studio e la Cura dei Tumori (IRST) "Dino Amadori" IRCCS, Meldola, Italy. [5] Molecular Biology Core Facility, Cancer Research UK Manchester Institute, The University of Manchester, Alderley Park, Macclesfield, Cheshire, UK. [6] Manchester Cancer Research Centre Biobank, The Christie NHS Foundation Trust, Manchester, UK. ✉email: richard.marais@cruk.manchester.ac.uk

mmunotherapy with programmed death 1 (PD1) checkpoint blockade (CPB) is increasingly utilized to treat solid tumors in the metastatic setting, where the response rate is 20–55%[1], and also as an adjuvant or neoadjuvant approach for locally advanced disease. Unfortunately, the identification of the patients who will benefit from treatment remains an unmet need, so to refine patient care predictive biomarkers are required to identify metastatic patients with better chances of responding to CPB. Moreover, in the adjuvant setting, tools are needed to select patients at lower risk of disease recurrence who have been cured by surgery so that they can be spared the risk of CPB-associated toxicity. The first biomarker strategies to emerge focused on the quantification of the tumor mutational burden and the expression of PD1 ligand (PD-L1) in the tumor microenvironment, but additional tools are needed to improve accuracy and the distinction between prognostic and predictive biomarkers[2,3].

The highly variable complementarity determining region 3 (CDR3) of the beta chain of the T cell receptor (TCR) is unique to individual T cell clones and can therefore be used to monitor the dynamics of T cell repertoire responses to CPB[4]. We recently reported that immune checkpoint inhibition with PD1 CBP can increase the clonality or the diversity of the TCR in peripheral T cells after 3 weeks of treatment, and importantly this bifurcated reaction only occurs in patients who respond to treatment[5]. This approach therefore allows patient responses to be monitored using minimally invasive liquid biopsies early during treatment.

Here we examine whether the TCR repertoire metrics in tumor-infiltrating T lymphocytes (TIL/Tc) can also identify which patients will benefit from CPB prior to commencement of therapy.

## Results

**Pre-treatment TIL/Tc clonality is predictive for CPB benefit.** To determine how the TCR repertoire of TIL/Tc influenced PD1 CPB outcome, we analyzed pre-treatment biopsies from 16 metastatic melanoma patients who received anti-PD1 agents in Manchester, Padova, and Meldola (Table 1). There were no differences in total TIL/Tc numbers, the number of TIL/Tc clones, or the diversity and clonality of the TCR when comparing lymph-node tumors and tumors from extranodal sites (Supplementary Fig. 1a–d), suggesting that the lymph-node microenvironment did not affect tumor infiltration by T cells. We also did not find significant correlation between patient overall survival (OS) and baseline peripheral serum lactic dehydrogenase levels, total number of TIL/Tc, number of TIL/Tc clones, TCR diversity, the number of non-synonymous single-nucleotide variants (SNVs), or PD-L1 staining (Supplementary Table 1). However, we did

find better OS and reduced death hazard (Cox regression $P = 0.0401$, hazard ratio $= 4.8 \times 10^{-14}$, C-index $= 0.88$) correlated to high TCR clonality in these pre-treatment tumor samples (Fig. 1A, B). We validated our findings using published data from 106 metastatic melanoma patients biopsied prior to treatment with PD1 or sequential PD1/CTLA4 blockade[6,7]. Here again, better OS correlated with higher TIL/Tc TCR clonality ($P = 0.0225$, external validation C-index $= 0.582$, 95% confidence interval $= 0.501$–$0.664$; Fig. 1C, D).

To ensure that OS was not confounded by subsequent treatments, we also tested the correlation between TIL/Tc TCR clonality and PD1 CPB response. In two independent metastatic melanoma patient cohorts[8,9], we found that pre-treatment TIL/Tc TCR clonality was directly associated with the likelihood of response for metastatic palliative and neoadjuvant PD1 CPB[10] (Fig. 1E–H, area under the curve [AUC] of the receiver operating curve [ROC] = 0.89 and 0.81and Fig. 1I, J, AUC of the ROC = 0.74, respectively), and although the numbers were too small to draw conclusions, only patients with TIL/Tc TCR clonality above the median have recurred at the time of analysis (Supplementary Fig. 2a). Intriguingly, the analysis of TIL/Tc TCR clonality in the on-treatment biopsy did not increase the performance of the prediction for response (Supplementary Fig. 2b, c, AUC of ROC = 0.74).

Notably, we observed consistency of our findings across the two platforms (TCR reconstruction from RNA sequencing (RNA-Seq) and targeted TCR sequencing) used to reconstruct the TCR in these melanoma cohorts and also with no significant difference in the overall TIL/Tc clonality, although TCR reconstruction from the RNA-Seq data yielded overall fewer unique TCR clonotypes than from the targeted TCR sequencing (Supplementary Fig. 3a, b). The cumulative CDR3 length was also comparable in the samples analyzed with the two platforms (Supplementary Fig. 3c), and we did not observe significant skewing of V gene representation (Supplementary Fig. 4a, b), nor striking differences in the similarity scores across samples, since these were largely unrelated in the two series (Supplementary Fig. 5a, b). Thus, high TCR clonality in the pre-treatment TIL/Tc population was predictive for PD1 CPB activity and efficacy across multiple series and using two methods to identify the TCR sequences.

**TIL/Tc diversity is prognostic for OS of melanoma patients in the absence of anti-PD1 inhibitors.** To test whether high TIL/Tc TCR clonality also correlated to prognosis in melanoma patients who did not receive PD1 CPB, we analyzed the TIL/Tc TCR repertoire in The Cancer Genome Atlas (TCGA) melanoma cohort[11], because the data-lock for these samples predated approval of anti-PD1 inhibitors. As expected, in multivariate analysis, age, clinical stage, and prevalence of single base substitution signature 7 (SBS7v2)[12,13] were prognostic for OS ($P < 0.001$; Fig. 2A and Supplementary Table 2). Notably, whereas TIL/Tc TCR clonality was not prognostic for survival in this cohort (Supplementary Table 2), high TIL/Tc TCR diversity was associated with better OS (Fig. 2B). Thus, high TCR diversity in pre-treatment TIL/Tc was prognostic for OS in melanoma patients who did not receive anti-PD1 treatments, and the addition of TCR diversity to the standard clinical covariates significantly increased the prognostic C-index (from 0.646 to 0.693, analysis of variance $P < 0.001$). To quantify cumulative melanoma risk based on these findings, we developed a nomogram to assist clinical decisions (Fig. 2C) and provide an example of the use of this tool (Supplementary Fig. 6).

**TIL/Tc diversity is prognostic for OS across several cancer histotypes.** Since PD1 CPB is now standard of care for melanoma

**Table 1 Training cohort clinical characteristics.**

|  | N (%) | Median (range) |
|---|---|---|
| Age (years) |  | 68 (28–83.7) |
| Sex |  |  |
| Male | 7 (44) |  |
| Female | 9 (56) |  |
| Drug received |  |  |
| Nivolumab | 4 (25) |  |
| Pembrolizumab | 12 (75) |  |
| Stage |  |  |
| Unresectable stage III | 2 (12) |  |
| M1a | 6 (38) |  |
| M1b | 1 (6) |  |
| M1c | 7 (44) |  |
| Lactic dehydrogenase (U/L) |  | 287 (51–822) |

The table shows the number of patients with the given characteristic with the percentage in parentheses or the median value for the variable with the range in parentheses.

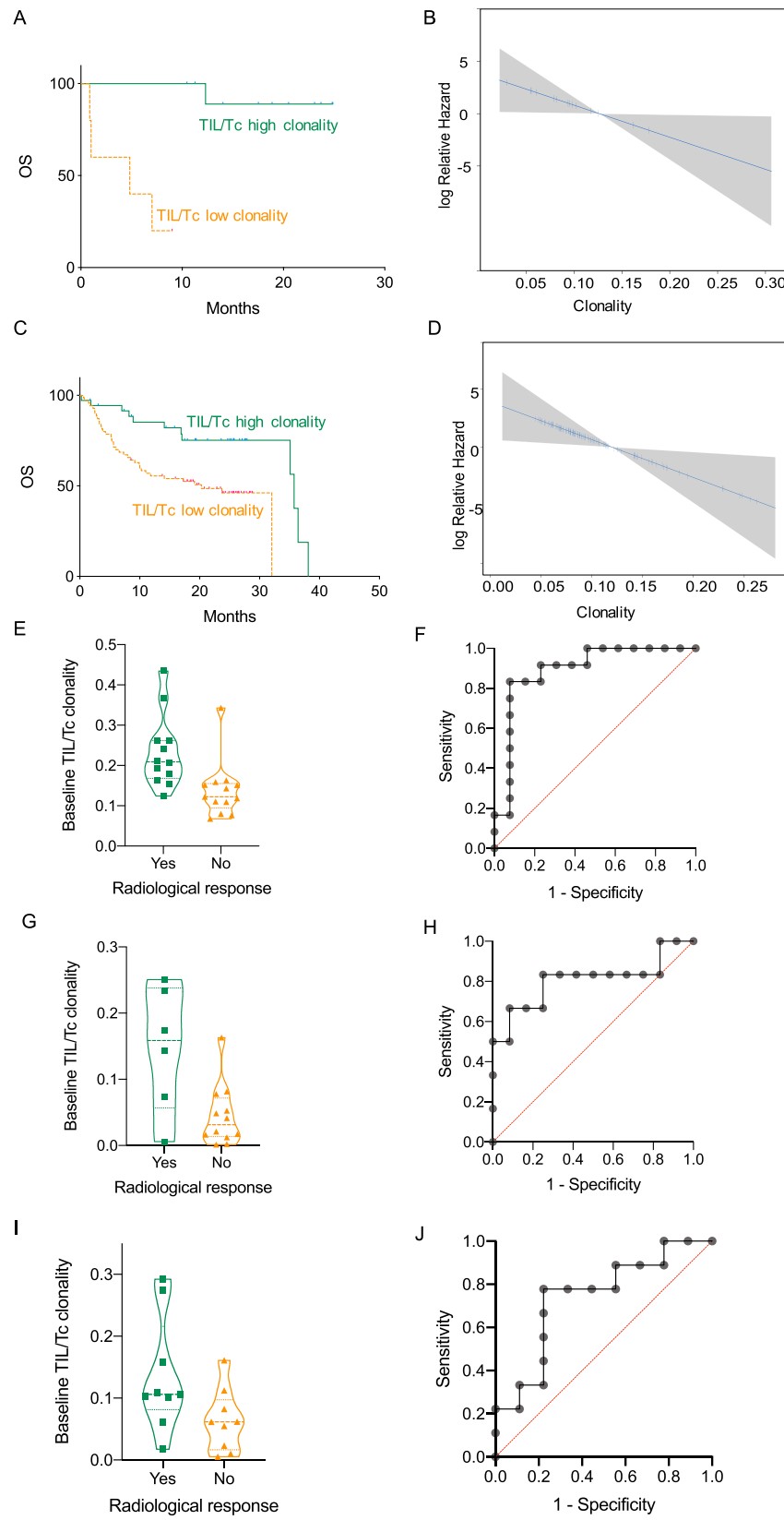

in the adjuvant and metastatic settings, prospective analysis of anti-PD1 naive melanoma patients who will not receive CPB in their clinical history is no longer feasible, so we examined TIL/Tc TCR diversity and clonality in other TCGA cancer cohorts. This also allowed us to assess whether the prognostic value of pre-treatment TIL/Tc is unique to melanoma or is shared by other

cancers. We focused on the cancers with an average TIL/Tc count in the highest quartile (>317.5), which includes breast cancer (BRCA), melanoma (SKCM), squamous lung carcinoma (LUSC), lung adenoma (LUAD), thymoma (THYM), clear cell renal cancer (KIRC), and testicular cancer (TGCT) (Fig. 3A and Supplementary Tables 3–7). As in previous studies[14], we restricted

**Fig. 1 TIL/Tc TCR repertoire in melanoma biopsies is predictive for overall survival and response to PD1 CPB. A** Survival curves for our metastatic melanoma training cohort of patients treated with anti-PD1 drugs with high (green) or low (orange) pre-treatment TIL/Tc clonality ($n = 16$, cut-off $= 0.06$, log-rank $P = 0.0003$), **B** predictive effect of TIL/Tc clonality on the relative hazard for death in the same cohort as **A**. Y-axis: log of relative hazard; a hazard ratio (HR) of 1 corresponds to 0, upper values correspond to HR > 1, and lower values correspond to HR < 1. The blue curves represent the HR function and "Rug plots" on curves show the density of the predictor (univariate Cox regression $P = 0.0193$); pointwise 95% confidence bands (shadowed area) are also shown. **C** Survival curves for the metastatic melanoma validation cohort[6, 7] of patients treated with anti-PD1 drugs with high (green) or low (orange) pre-treatment TIL/Tc clonality ($n = 106$, cut-off $= 0.10$, log-rank $P = 0.0039$), **D** predictive effect of TIL/Tc clonality on the relative hazard of death in the same cohort (univariate Cox regression $P = 0.0225$). **E** Violin plots of the pre-treatment TIL/Tc clonality distribution in patients who achieved radiological response (green, $n = 12$, median $= 0.23$, SD $= 0.09$) and progressed (orange, $n = 13$, median $= 0.14$, SD $= 0.07$) to treatment with PD1 CPB in a metastatic melanoma cohort[9] (simple logistic regression log-likelihood ratio for association with probability of response $= 8.6$, $P = 0.0033$, $n = 25$) and **F** receiver operating curve (ROC) of the linear regression response prediction (area under the curve $= 0.89$). **G** Violin plots of the pre-treatment TIL/Tc clonality distribution in patients who achieved radiological response (green, $n = 6$, median $= 0.15$, SD $= 0.09$) and progressed (orange, $n = 12$, median $= 0.04$, SD $= 0.05$) to treatment with PD1 CPB in a metastatic melanoma cohort[8] (simple logistic regression log-likelihood ratio for association with probability of response $= 7.2$, $P = 0.007$, $n = 18$) and **H** receiver operating curve (ROC) of the linear regression response prediction (area under the curve $= 0.81$). **I** Violin plots of the pre-treatment TIL/Tc clonality distribution in patients who achieved radiological response (green, $n = 9$, median $= 0.14$, SD $= 0.09$) and progressed (orange, $n = 9$, median $= 0.06$, SD $= 0.05$) to treatment with neoadjuvant PD1 CPB in an advanced melanoma cohort[10] (simple logistic regression log-likelihood ratio for association with probability of response $= 4.5$, $P = 0.0340$, $n = 18$) and **J** receiver operating curve (ROC) of the linear regression response prediction (area under the curve $= 0.74$). Analyses are two-sided; TIL/Tc clonality and diversity are retained as continuous variables in the regression analyses; $n$ is single patient; single green dots represent single patients; horizontal dotted lines in the violin plots represent median and SD; SD $=$ standard deviation. Source data are provided as a Source data file.

our LUSC analysis to patients with low cigarette use, and due to insufficient OS events[15], we used progression-free interval as the TGCT endpoint. Also, due to the small number of events for patients with TCR data, we could not perform regression analysis for THYM[15]. Despite these restrictions, we found that high TCR diversity in the TIL/Tc was associated with improved survival in all of these cancers (Fig. 3B–L) while clonality was not, although it had borderline significance for OS in KIRC (Supplementary Table 7).

## Discussion

Our data shows that, as in melanoma, in various other cancers TIL/Tc diversity is prognostic for OS in the absence of PD1 blockade, whereas pre-treatment TIL/Tc clonality is predictive of response to anti-PD1 treatment. This is consistent with observations that on-treatment TIL/Tc clonality anticipates radiological response to immunotherapy in melanoma[8,16]. Our results suggest that high TCR diversity in TIL/Tc identifies patients whose immune system achieves durable tumor control without anti-PD1 therapy, whereas high TIL/Tc clonality identifies which patients will mount an effective anti-PD1-induced immune response.

The reconstruction of the TCR using cancer transcriptomic datasets is a powerful approach in defining the T cell repertoire in solid tumors, but here we show that targeted TCR sequencing approaches identify more unique TCR clonotypes than reconstruction from RNA-Seq data and consequently targeted sequencing could provide a more robust resolution when analyzing samples presenting limited numbers of TIL/Tc. Despite this, we showed consistent results with the two platforms, supporting our hypothesis that TIL/Tc repertoire analysis provides new ways to explore delivery of personalized immunotherapy, although we expect more sensitive and accurate approaches to eventually replace existing state-of-the-art solutions.

With adjuvant PD1 CPB already approved in stage III melanoma and currently being trialed in stage II melanoma (NCT03553836), the need for predictive and prognostic stratification tools for CPB is becoming more pressing. This is particularly important for patients with intermediate-risk disease where the odds of toxicity could outweigh the benefit of adjuvant immunotherapy and where the TIL/Tc repertoire could provide a much-needed reliable biomarker for patient selection.

More studies are needed to determine why high TIL/Tc clonality is predictive of CPB benefit, whereas TIL/Tc diversity is prognostic in the absence of CPB, but our findings nevertheless have important clinical implications because of their potential to contribute to the development of personalized therapeutic strategies through the identification of the patients who could better benefit from treatments.

## Methods

**Patients**. The training cohort patient samples do not involve clinical trials/clinical trial-associated data, as patient samples were collected prior to standard-of-care treatment. Biopsy pre-treatment samples from Manchester patients were prospectively collected under the Manchester Cancer Research Centre (MCRC) Biobank ethics application #18/NW/0092; the study was approved by MCRC Biobank Access Committee application 13_RIMA_01. Biopsy pre-treatment samples and clinical information from Padova patients were collected under the University of Padova Department of Surgery, Oncology and Gastroenterology ethics application 04/03/2002 protocol #448 and Veneto Oncology Institute ethics approval 09/04/2018 protocol #006264. Biopsy pre-treatment samples from Meldola patients were collected under ethics application #5483/2018 of Comitato Etico della Romagna. All patients gave written informed consent to the use of the samples for research purposes. Clinical endpoints: date of death was obtained from clinical records (pre-CPB training cohort), the original publications (pre-CPB validation cohorts), and TCGA clinical information files. Radiological response information was obtained from the original publications.

**RNA sequencing**. RNA was extracted from pre-treatment human fresh frozen tumor samples using the AllPrep DNA/RNA Kit (Qiagen, Manchester, UK) according to the manufacturer's instructions: briefly, tissue samples were first lysed and homogenized in a highly denaturing guanidineisothiocyanate-containing buffer to inactivate DNases and RNases and ensure isolation of intact DNA and RNA; the lysates were then passed through an AllPrep DNA membrane that, in combination with the high-salt buffer, binded the genomic DNA that was then eluted; ethanol was added to the flow-throughs from the AllPrep DNA membrane to provide appropriate binding conditions for RNA, and the samples were then applied to an RNeasy membrane; total RNA binded to the membranes and contaminants were washed away; high-quality RNA was then eluted in 45–70 µl water. Indexed PolyA libraries were prepared using 200 ng of total RNA and 14 cycles of amplification with the Agilent SureSelect Strand Specific RNA Library Prep Kit for Illumina Sequencing (Agilent, G9691B, Santa Clara, CA, US). Libraries were quantified by quantitative PCR (qPCR) using the KAPA Library Quantification Kit for Illumina platforms (Kapa Biosystems Inc., KK4873, Wilmington, MA, US). Paired-end 100 bp sequencing was carried out by clustering 15 pM of pooled libraries on the cBot and sequenced on the Illumina HiSeq 2500 in high output mode using TruSeq SBS V3 chemistry (Illumina Inc., San Diego, CA, US); average of all samples was 68 million pass-filter (PF) reads (each end). The primers were all supplied as part of the kits that are listed. After removing adapters using Cutadapt (v1.14) and trimming poor quality base calls using Trimmomatic (v0.36)[17], the human reads were aligned to GRCh37 (release 75) using STAR (v2.5.1) aligner[18].

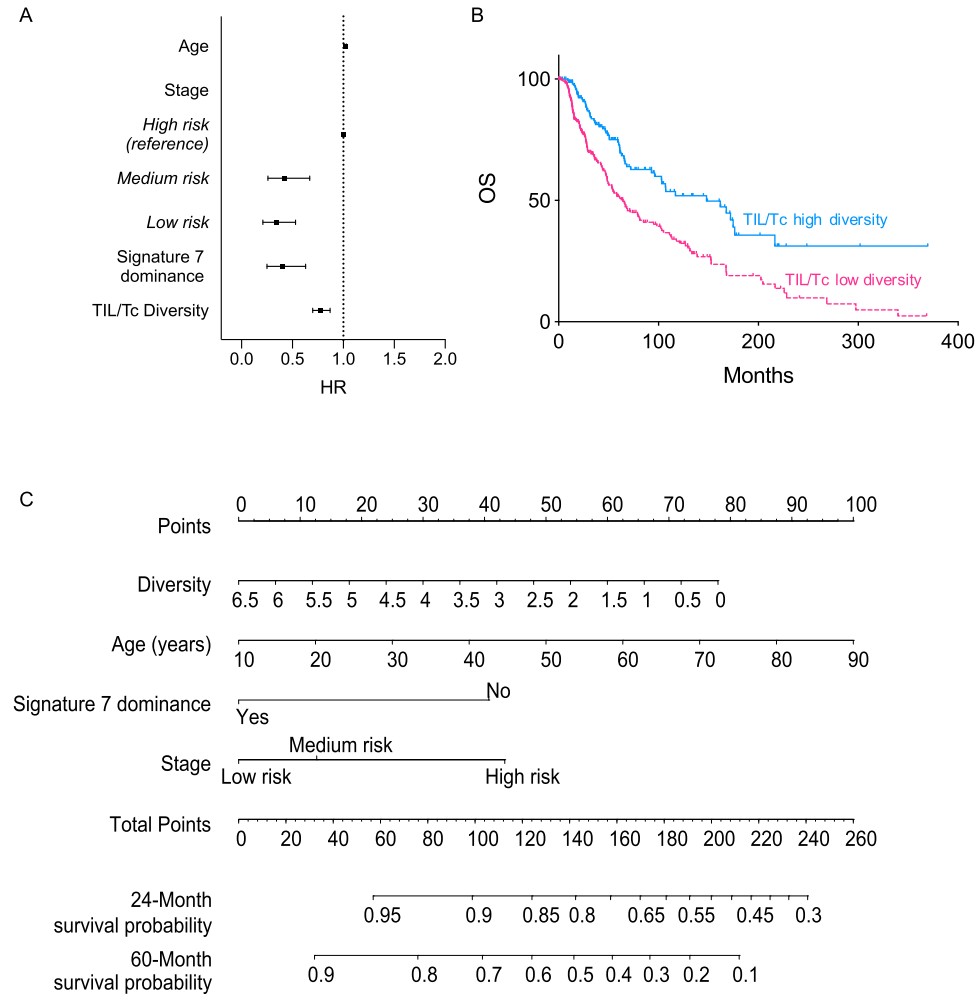

**Fig. 2 TIL/Tc TCR repertoire in melanoma biopsies is prognostic for overall survival in absence of PD1 CPB.** Analyses of TCGA skin melanoma cohort (n = 412): **A** forest plot showing HR and 95% CI for significant covariates retained in the multivariate Cox regression prognostic model calculated using the fast-backward method and the Akaike Information Criterion as a stopping rule (events = 147, model for OS global P < 0.001, concordance index = 0.71); **B** survival curves for patients with high (blue) or low (pink) TIL/Tc diversity (cut-off = 3.49, log-rank P < 0.0001); **C** nomogram tailored on the final model with the significant prognostic factors from **A**; the sum of the prognostic factor values corresponds to the survival probability at 24 and 60 months, an example of the nomogram use is shown in Supplementary Fig. 6. Analyses are two-sided; TIL/Tc clonality and diversity are retained as continuous variables in the regression analyses; n is single patient, OS is overall survival, HR is hazard ratio, CI is confidence interval; signature 7 is single base substitution signature 7 version 2[12, 13]. Source data are provided as a Source data file.

**TCR analysis.** CDR3 TCR sequence data generated by Bo Li et al.[11] for the 29 TCGA cohorts were kindly made available by the authors; the clinical data for the cohorts were downloaded from cBioportal in February 2018. CDR3 TCR sequences were inferred from RNA Seq data from pre-treatment biopsies for the training cohort samples using ImReP[19]; inclusion criterion for downstream analyses was minimum four TCR sequences identified. Clonality was calculated with the function *clonality* from *LymphoSeq* R package; heatmaps were generated using *LymphoSeq*, *pheatmap*, and *ggplot2* R packages. The diversity was calculated using Renyi index ($\alpha = 1$) as per Spreafico et al.[20].

Whole-exome sequencing snap-frozen tumor tissue was manually dissected by sectioning (25-µm thick), and DNA was extracted from sections with an estimated tumor cell percentage of at least 80% using the AllPrep DNA/RNA Kit (Qiagen) according to the manufacturer's instructions, as per description above. Germline DNA was isolated from patients' blood. DNA quantity was assessed using a Qubit® 2.0 Fluorometer (Life Technologies).

One microgram of genomic DNA was sheared using a Covaris S2 ultrasonicator (Covaris, Inc.). Multiplexed libraries were prepared using the SureSelectXT Target Enrichment System for Illumina Paired-End Sequencing and the SureSelect Human All Exon V6 Capture Library (Agilent, G9641B/5190-8864).

Libraries were quantified by qPCR using the KAPA Library Quantification Kit for Illumina platforms (Kapa Biosystems, Inc., KK4873). Paired-end 100 bp sequencing was carried out by clustering 14 pM of pooled libraries on the cBot and sequenced on the Illumina HiSeq 2500 in high output mode using TruSeq SBS V3 chemistry (Illumina, Inc.); average PF reads (each end) was 126 million for tumor samples and 67 million for germline samples. The primers were all supplied as part

of the kits that are listed. After removing adapters using Cutadapt (v1.14) and trimming poor quality base calls using Trimmomatic (v0.36), the reads were aligned to the GRCh37 (release 75) human genome using BWA aligner (v0.7.7). The PCR duplicate reads were filtered using Picard (v1.96), and the base quality score recalibration and local INDEL realignments were performed using GATKtools (v3.1). Using tumor–normal pairs, SNVs were identified using MuTect[21] (v1.1.7). Variant Effect Predictor (Ensembl version 73/84) was used to annotate the mutations. Known variants present in dbSNP were excluded.

**Mutational signature.** Mutational signatures for TCGA SKCM cohort were determined by fitting somatic SNVs with tri-nucleotide context to the 30 COSMIC mutational signatures using deconstructSigs[12] package using default parameters. Signatures with contribution weights <6% were excluded.

**PD-L1 quantification in pre-treatment biopsies.** Pre-treatment biopsy samples were fixed in 10% neutral buffered formalin (Sigma-Aldrich), processed, and embedded in paraffin wax. Samples were sectioned (4 µm) and stained with hematoxylin and eosin using standard protocols. Formalin-fixed paraffin-embedded sections were deparaffinized using xylene and rehydrated passing through a series of graded ethanol to distilled water steps. Heat-induced epitope retrieval (125 °C, 60 s; 90 °C, 10 s) was performed using a Pascal pressure chamber (Dako) with Dako target retrieval solution pH 6 (S203130, Agilent Technologies, Santa Clara, California, US). Sections were then processed and stained with anti-PD-L1 22C3 PharmDx (Agilent Technologies, Santa Clara, CA, US) as per the

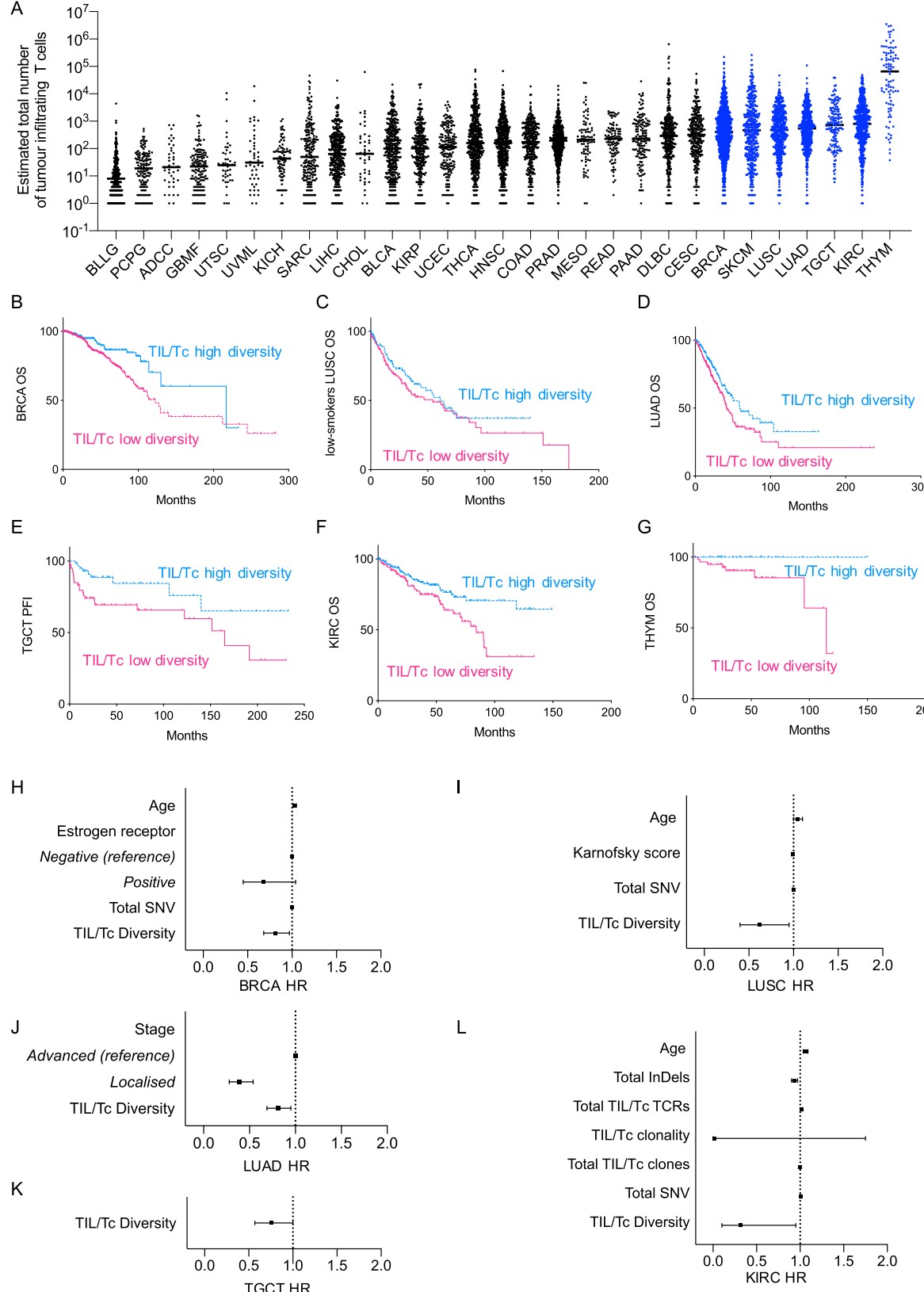

manufacturer's instructions: briefly, after peroxidase block for 5 min, the sections were incubated with the 22C3 antibody using a concentration of 1:50 for 60 min at room temperature and then incubated for 30 min at room temperature with the linker antibody provided with the kit and specific to the host species of the primary antibody; finally, the sections were incubated with a ready-to-use visualization reagent provided with the kit and consisting of secondary antibody molecules and horseradish peroxidase molecules coupled to a dextran polymer backbone. Sections

were then quantified by a clinical pathologist according to good clinical practice guidelines for approved diagnostic PD-L1 quantification.

**Statistics and reproducibility**. All tests were two-sided and $P$ values < 0.05 were retained as significant. Unless otherwise specified, multivariate Cox regression was used to calculate the hazard of death; fast-backward method was applied to select

**Fig. 3 TIL/Tc TCR diversity is prognostic across several cancer histotypes. A–L** Analyses of the TCGA cancer cohorts. **A** Estimated total number of tumor-infiltrating T cells (TIL/Tc) in the cancer cohorts; the cohorts above the upper quartile of TIL/Tc are highlighted in blue; **B–G** survival curves for the cancer cohorts with abundant TIL/Tc from **A** of patients with high (blue) or low (pink) TIL diversity: **B** BRCA ($n = 908$, cut-off $= 3.34$, OS log-rank $P = 0.0029$), **C** LUSC patients with cigarette use < median ($n = 238$, cut-off $= 3.17$, OS log-rank $P = 0.2849$), **D** LUAD ($n = 481$, cut-off $= 3.53$, OS log-rank $P = 0.0204$), **E** TGCT ($n = 148$, cut-off $= 3.29$, DFI log-rank $P = 0.0124$), **F** KIRK ($n = 322$, cut-off $= 3.26$, OS log-rank $P = 0.0088$), **G** THYM ($n = 92$, cut-off $= 5.93$, OS log-rank $P = 0.0306$); **H–L** forest plots showing the HR and 95% CI for the significant covariates retained in the multivariate Cox regression prognostic model calculated using the fast-backward method and the Akaike Information Criterion as a stopping rule for the cancer cohorts with abundant TIL/Tc from A; TIL/Tc metrics were analyzed as continuous variables: **H** BRCA (events $= 113$, model for OS global $P < 0.001$, concordance index $= 0.66$), **I** LUSC patients with cigarette use < median (events $= 31$, model for OS global P $= 0.0096$, concordance index $= 0.69$), **J** LUAD (events $= 166$, model for OS global $P < 0.001$, concordance index $= 0.63$), **K** TGCT (events $= 31$, model for PFS $P = 0.0637$, concordance index $= 0.63$), **L** KIRK (events $= 53$, model for OS global $P < 0.001$, concordance index $= 0.74$). Analyses are two-sided and TIL/Tc diversity is retained as continuous variable in the regression analyses; $n$ is single patient, OS is overall survival, DFI is disease-free-interval, PFS is progression-free survival, HR is hazard ratio, CI is confidence interval. (BLLG brain lower grade glioma, $n = 289$; PCPG pheochromocytoma and paraganglioma, $n = 142$; ADCC adrenocortical carcinoma, $n = 40$; GBMF glioblastoma multiforme, $n = 140$; UTSC uterine carcinosarcoma, $n = 45$; UVML uveal melanoma, $n = 47$; KICH kidney chromophobe, $n = 82$; SARC sarcoma, $n = 218$; LIHC liver hepatocellular carcinoma, $n = 383$; CHOL cholangiocarcinoma, $n = 42$; BLCA bladder urothelial carcinoma, $n = 380$; KIRP kidney renal papillary cell carcinoma, $n = 295$; UCEC uterine corpus endometrial carcinoma, $n = 184$; THCA thyroid carcinoma, $n = 518$; HNSC head and neck squamous cell carcinoma, $n = 524$; COAD colon adenocarcinoma, $n = 325$; PRAD prostate adenocarcinoma, $n = 536$; MESO mesothelioma, $n = 83$; READ rectum adenocarcinoma, $n = 100$; PAAD pancreatic adenocarcinoma, $n = 145$; DLBC diffuse large B cell lymphoma, $n = 331$; CESC cervical squamous cell carcinoma and endocervical adenocarcinoma, $n = 296$; BRCA breast carcinoma, $n = 1160$; SKCM skin cutaneous melanoma, $n = 426$; LUSC lung squamous carcinoma, $n = 528$; LUAD lung adenocarcinoma, $n = 585$; TGCT testicular germinal cell tumors, $n = 153$; KIRC kidney renal clear cell carcinoma, $n = 597$; THYM thymoma, $n = 93$). Source data are provided as a Source data file.

the covariates retained in the multivariate models (*fastbw* function in *rms* R package, type = "*individual*"). The Akaike Information Criterion [AIC] as a stopping rule was used to select the covariates in the final model in order to weight the probability of both significance and prediction strength. The model performance was quantified with C-indexes, calculated after validation with 200 bootstraps (*validate* in rms R package, rule = "*aic*") and compared with analysis of variance. A nomogram (rms R package) was tailored on the final regression model for the melanoma TCGA cohort; the total number of points derived by specifying the covariate values was used to calculate the expected survival probabilities at 24 and 60 months; to proceed with the nomogram design, the missing values in the final regression model were estimated with multiple imputations using additive regression, bootstrapping, and predictive matching; a correction on the estimation procedure was based on 200 multiple imputations. Cox–Snell residuals were used to verify the proportional hazard hypothesis (with a P value >0.05 confirming the hypothesis for all covariates retained in the multivariate models with the exception of melanoma stage and breast cancer estrogen receptor status, which were time dependent). Kaplan–Meier method with log-rank test was used to plot survival data and patients' allocation to groups were based on the biomarker cut-off determined with *OptimalCutpoints* R package, with the exception that the TCR metrics were retained as continuous variables for all the regression analyses to avoid cut-off artifacts and selection bias. Simple logistic regression was applied to calculate the log-likelihood ratio of radiological response to anti-PD1 therapy according to the clonality of baseline biopsy TIL/Tc; the AUC of the ROC was calculated for the parameters sensitivity and $1 -$ specificity. We used the "rule of the thumb" to determine the maximum number of covariates to use in the regression models[22]. Samples with less than four TCR sequences were outbound for the algorithm to calculate Renyi index and were excluded from the analysis. TCGA samples with incomplete clinical annotation and no survival information were included in the TCR analysis shown in Fig. 3A but were excluded from the survival and Cox regression analyses in Fig. 2 and Fig. 3B–L. The investigators were blinded during experiments; outcome assessment was performed after experiments. Analyses were performed with GraphPad Prism version 7 (GraphPad Software, La Jolla, CA, USA) or R (v. 3.6.3, The R Foundation for Statistical Computing, Vienna, Austria). Data reporting follows REMARK guidelines (REporting recommendations for tumor MARKer prognostic studies).

**Reporting summary**. Further information on research design is available in the Nature Research Reporting Summary linked to this article.

## Data availability

Sample metadata file are available as supplementary material. The RNA-Seq data for the pre-treatment training cohort have been deposited in EGA under the accession code EGAS00001005201. TCR sequencing data for the TCGA cohorts were kindly made available by Bo Li et al.[11]. Treatment-naive melanoma biopsy TCR sequencing data of the pre-treatment validation cohort reanalyzed here for OS were downloaded from https://clients.adaptivebiotech.com/pub/weber-2018-cir (https://doi.org/10.21417/EY2019CIR) and https://github.com/riazn/bms038_analysis. Pre-treatment melanoma biopsy TCR sequencing data of the response analysis cohorts[9,10,16] reanalyzed here were downloaded from https://clients.adaptivebiotech.com/pub/tumeh-2014-nature, manuscript

supplementary material of the original manuscripts, and EGAS00001003178 EGA study accession dataset EGAD00010001608 (only patients with matched baseline and week 3 tumor samples were included, to allow for AUC comparison of the ROC). The progression-free survival data for the 11 patients in the neoadjuvant cohort who were studied for progression-free survival (Supplementary Fig. 2a) were kindly made available by the original manuscript authors. In-del data for TCGA clear cell renal carcinoma cohort were kindly made available by Turajlic et al.[14]. Source data are provided with this paper.

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

## Acknowledgements

We are grateful to the patients who participated to this study and to their families. We thank International Neoadjuvant Melanoma Consortium (INMC) for providing the progression-free survival data for the advanced melanoma neoadjuvant cohort. We thank Professor Judi Allen and the Molecular Oncology Group for their advice. This work was supported by CRUK (A27412 and A22902), the Harry J Lloyd Charitable Trust (Career Development Award for SV), and the Wellcome Trust (100282/Z/12/Z). The role of the MCRC Biobank is to distribute samples and therefore cannot endorse studies performed or the interpretation of results.

## Author contributions

Conception and design: S.V. and R.M. Development of methodology: S.V. and P.A.M. Acquisition of data (managed patients, provided facilities, performed bioinformatics analyses, performed experiments etc.): S.V., E.G., P.A.M., L.G.C., F.D.R., A.G., S.M., J.W., P.L.C., N.D., R.M. Manuscript writing: S.V., N.D., and R.M. Correction and approval of the manuscript: all authors.

## Competing interests

R.M. is an expert witness for Pfizer, and as a former Institute of Cancer Research (London) employee, he may benefit from commercialized programs. P.L. serves as a paid advisor/speaker for Bristol-Myers Squibb, Merck Sharp and Dohme, Roche, Novartis, Amgen, Pierre Fabre, Nektar, and Melagenix. P.L. reports travel support from Bristol-Myers Squibb and Merck Sharp and Dohme and receives research support from Bristol-Myers Squibb. A.G. received honoraria and consultancy fees from BMS and Novartis. L.G.C. served as a paid advisor for Igea. F.D.R. serves as a paid advisor for MSD and had travel support from BMS and Novartis and honoraria from BMS. Other authors declare no competing interests.
