## [Peer Review File · Nature Communications]

Reviewers' Comments:

Reviewer #1:

Remarks to the Author:

In this study Valpione et al report that the TCR repertoire has prognostic value for cancer survival. In particular they have shown that T cell clonality is predictive of survival in a cohort of 16 patients treated with anti-PD1. They validated this finding by analysing a separate cohort of 106 metastatic melanoma patients published by Riaz et al. 2017 and Yosko et al. 2019.

The advance from the group's previous work is that they have evaluated the predictive power of TCR repertoire in pre-treatment melanoma patient samples. In addition, they analysed the predictive power of the TCR repertoire for TCGA cancer datasets that are on the top of the spectrum of T cell infiltration.

Overall, I believe this work is convincing and scientifically sound. It advances knowledge in the field of tumour immunology, it is clinically relevant and will be informative for scientists in the field even though it lacks a more comprehensive insight into the underlying immune response mechanisms.

While the results are interesting, there are concerns with the manuscript in its current form:

1. The paper is not structured into Introduction, Results and Discussion and the data have not been discussed sufficiently.
2. A more detailed presentation of the TCR analysis should be included i.e. average clonotype sizes, spectratypes, VJ usage profiles, read counts etc.
3. The high and low thresholds applied for the survival curve analyses need to be explained more explicitly in Methods.
4. In Figure 1G, the nomogram needs more detail on its construction and interpretation in the main text.
5. The depth of sequencing data should be included in the relevant section of Methods.

Reviewer #2:

Remarks to the Author:

In their manuscript, Valpione et al. examine whether clonality and diversity of the TCR repertoire of tumor infiltrating lymphocytes can serve as a prognostic and predictive biomarker in the cancer setting, with a particular interest in metastatic melanoma patients receiving PD1 blockade therapy. The authors analyze biopsies of a cohort metastatic melanoma patients as well as a number of TCGA datasets to expand the scope of the study. The authors present convincing data showing that TCR clonality can predict the response to anti-PD1 in metastatic melanoma patients and, on a larger scale, how TCR diversity can predict overall survival independently of PD1 treatment in numerous cancers. Such findings may impact which patients receive PD1 based immunotherapy.

Comments

- 1) Line 44. Define LDH when first used.
- 2) Line 58-60: Since clonality is discussed In the statement, "Notably, whereas TIL/Tc... with better OS (Fig. 1e,f).", either plot clonality in Figure 1e or include Extended Data Table 2 in the figure reference.
- 3) Figure 1a-d: It would appear that in both melanoma cohorts, more patients tend to have low clonality. Is such a distribution to be expected? Is it then correct to deduce that a majority of metastatic melanoma patients should not receive PD1 blockade despite this being the standard treatment? How has anti-PD1 become the standard if this is the case?
- 4) Figure 1f: Please reconcile the color indicated in the figure legend with the color in the graph since the graph appears pink and not purple.
- 5) A comparative analysis of the pre-PD1 versus post-PD1 TCR repertoire (if available) would be of

interest and could provide some insight.

Reviewer #3:

Remarks to the Author:

Valpione and colleagues investigate the TCR repertoire in melanoma biopsies from newly diagnosed patients and attempt to find correlates based on the hypothesis that the TCR repertoire of TIL could determine the disease course (clinical outcome). 16 patients recruited from 3 centers (Manchester, Padova, Meldola) received initial systemic therapy with anti-PD1 agents. The authors state that there were no differences in TIL numbers, TIL clones/clonotypes, TCR diversity, or TCR clonality (Extended data Fig 1). In addition, no correlation was found between OS and serum LDH, TIL numbers, TIL clones, TCR diversity, #SNVs, or PD-L1 expression levels (Extended data table 1). They did, however, find an association between OS and TCR clonality (Figure 1). Validation was assessed using datasets from Riaz (2017) and Yusko (2019) with melanoma patients (n=106) treated with either anti-PD1 or the combination anti-CTLA4/anti-PD1. The investigators then examined TCGA data base and found that TCR diversity (not clonality) as a prognostic marker correlated with OS for several cancers including BRCA, LUAD, TGCT, KIRK, THYM (Fig 2). The authors conclude: "high TIL/Tc clonality identifies which patients will mount an effective anti-PD1-induced immune response" and that this observation has important clinical implications.

This is an interesting observation; however, it is based on modest data with no mechanistic insight. Several major concerns are listed below.

1. The authors studied 16 patients with metastatic melanoma. Please describe the clinical, pathologic, radiographic, and demographic features of these individuals. At first glance, the sample size appears exceedingly small and if this association (OS and high TCR clonality) is true, it should be broadened to be truly representative of newly diagnosed patients to convince one that others can build upon these results. The limited information in the data summary is inadequate.

2. Data shown in Extended Data Figure 1 panel B reports the "total unique TCR sequences" for 6 data points from extranodal sites and 10 data points from nodal sites. The median number of unique TCR sequences in each group is 10 which is dramatically lower than expected based on the literature (and our own experience). For example, most tumor biopsies contain thousands or tens of thousands of unique clonotypes based on TCR VB CDR3 sequences obtained from melanoma patients (Pasetto CIR 2016; Riaz Cell 2017) or other histologies (Wu Nature 2020) using Adaptive Biotechnologies method of bulk sequencing gDNA or 10x Genomics sc mRNA platform. The low number of unique TCR clonotypes for each patient probably reflects a sampling bias or alternatively, a sequencing artifact due to their methodology. Finally, the authors should be more precise in their definition of a TCR clonotype and the primary sequencing data for all 16 subjects should be provided.

3. The rationale to include TCGA data (Figure 2) from other cancer types is puzzling and some would argue it is extraneous to the observation made in figure 1. Namely, TCR clonality at baseline is predictive for clinical benefit after anti-PD1 therapy as first-line therapy. It would appear logical to investigate non-melanoma patients with newly diagnosed cancer administered anti-PD1 (either on study or as standard of care) to provide confirmatory evidence. The most pressing question for investigators in the field concerns the specificity of the high frequency TCRs evident in some patients and this was not addressed in the current study.

Minor Points

1. A point worth mentioning relates to an assumption implicit in the authors conclusion that "an effective anti-PD1-induced immune response" somehow equates with improved OS. The data in Figure 1 concerns OS as a clinical endpoint. The authors show no data related to the quality, composition or magnitude of the T cell response after anti-PD1 treatment.

2. The authors appear to state that a prospective analysis of anti-PD1 naïve patients is no longer feasible since adjuvant administration is now standard of care. I would be cautious about this

conclusion as many stage II (and a small percentage of stage I) patients will not develop nodal regional disease but rather proceed to stage IV metastatic disease that can be biopsied to assess TCR clonality prior to first line anti-PD-1.

NCOMMS-20-30697-T: Responses to reviewers' comments.

We are grateful to the reviewers for the time spent on our manuscript. Their comments were both insightful and helpful, and in responding, the quality of the manuscript is much improved. Below we provide our point-by-point responses to each comment (for ease the page and line information refers to the tracked version of the revised manuscript).

REVIEWER COMMENTS

Reviewer #1 (Remarks to the Author):

General comment. *In this study Valpione et al report that the TCR repertoire has prognostic value for cancer survival. In particular they have shown that T cell clonality is predictive of survival in a cohort of 16 patients treated with anti-PD1. They validated this finding by analysing a separate cohort of 106 metastatic melanoma patients published by Riaz et al. 2017 and Yosko et al. 2019.*

The advance from the group's previous work is that they have evaluated the predictive power of TCR repertoire in pre-treatment melanoma patient samples. In addition, they analysed the predictive power of the TCR repertoire for TCGA cancer datasets that are on the top of the spectrum of T cell infiltration.

Overall, I believe this work is convincing and scientifically sound. It advances knowledge in the field of tumour immunology, it is clinically relevant and will be informative for scientists in the field even though it lacks a more comprehensive insight into the underlying immune response mechanisms.

Response. We thank Reviewer #1 for their appreciation of our work. We are delighted that the reviewer found our work to be convincing and scientifically sound and are particularly pleased that they concur that it advances knowledge in tumour immunology, is clinically relevant and informative.

Comment. *While the results are interesting, there are concerns with the manuscript in its current form:*

Comment 1. *The paper is not structured into Introduction, Results and Discussion and the data have not been discussed sufficiently.*

Response. We thank the reviewer for this and have now changed the format from a brief communication to that of an article, which has allowed us to discuss our data more comprehensively.

Comment 2. *A more detailed presentation of the TCR analysis should be included i.e. average clonotype sizes, spectratypes, VJ usage profiles, read counts etc.*

Response. Thank you for the suggestion. In response we have added new descriptive and comparative analyses of the TCR repertoire in the cohorts studied and include these in new Extended Data Figures 3, 4 and 5.

Comment 3. *The high and low thresholds applied for the survival curve analyses need to be explained more explicitly in Methods.*

Response. Our apologies for the lack of detail. In response, we explain in the Methods (page 22 line 395) the methodology applied to identify the grouping cut-off for the survival curves and we do however wish to highlight that, although the categorisation of TIL/TC TCR clonality and diversity was necessary for the Kaplan-Meier curves, to avoid the potential bias introduced by cut-offs, all regression analyses were performed retaining the TCR metrics as continuous variables: this is now better explained in the Methods (22 line 396).

Comment 4. *In Figure 1G, the nomogram needs more detail on its construction and interpretation in the main text.*

Response. We thank the reviewer for the suggestion, and in response we have added details about the nomogram construction and interpretation in the Methods section (page 21 line 381 and page 22 line 385) and have additionally added an example of its use in the Results (page 6 line 118, Extended Data Fig 6).

Comment 5. *The depth of sequencing data should be included in the relevant section of Methods.*

Response. Apologies for the omission; more details have now been added to the Methods (page 19 line 327 and 20 line 351).

Reviewer #2 (Remarks to the Author):

General comment. *In their manuscript, Valpione et al. examine whether clonality and diversity of the TCR repertoire of tumor infiltrating lymphocytes can serve as a prognostic and predictive biomarker in the cancer setting, with a particular interest in metastatic melanoma patients receiving PD1 blockade therapy. The authors analyze biopsies of a cohort metastatic melanoma patients as well as a number of TCGA datasets to expand the scope of the study. The authors present convincing data showing that TCR clonality can predict the response to anti-PD1 in metastatic melanoma patients and, on a larger scale, how TCR diversity can predict overall survival independently of PD1 treatment in numerous cancers. Such findings may impact which patients receive PD1 based immunotherapy.*

Response. We are delighted that the reviewer found our results on TCR clonality to be convincing and that they agree that our data can predict patient response to anti-PD1 in melanoma. We are pleased that they agree that we show that TCR diversity can predict overall survival in numerous cancers and that our findings could impact which patients will receive PD1-based immunotherapy.

Comments

Comment 1. *Line 44. Define LDH when first used.*

Response. We thank the reviewer for spotting this, and have added the definition (page 4 line 71).

Comment 2. *Line 58-60: Since clonality is discussed in the statement, “Notably, whereas TIL/Tc... with better OS (Fig. 1e,f).”, either plot clonality in Figure 1e or include Extended Data Table 2 in the figure reference.*

Response. We are grateful for the suggestion for spotting this, we have now included the Extended Data Table 2 reference in the figure legend.

Comment 3. *Figure 1a-d: It would appear that in both melanoma cohorts, more patients tend to have low clonality. Is such a distribution to be expected? Is it then correct to deduce that a majority of metastatic melanoma patients should not receive PD1 blockade despite*

this being the standard treatment? How has anti-PD1 become the standard if this is the case?

Response. We thank Reviewer #2 for this observation. In response, we have made it clearer in the text that although the categorisation of TIL/TC TCR clonality and diversity was necessary to generate the Kaplan-Meier curves, to avoid potential bias introduced by cut-offs all the regression analyses were performed retaining the TCR metrics as continuous covariates (page 22 line 396). The reviewer's observation also prompted us to comment in the Introduction that although anti-PD1 drugs have revolutionised the treatment of melanoma, only about 20-55% of treated patients benefit from these drugs (page 3 line 39).

Comment 4. *Figure 1f: Please reconcile the color indicated in the figure legend with the color in the graph since the graph appears pink and not purple.*

Response. Apologies for the mistake and thank you for highlighting it. It is now corrected (now in Figure 2b and 3b-g legends).

Comment 5. *A comparative analysis of the pre-PD1 versus post-PD1 TCR repertoire (if available) would be of interest and could provide some insight.*

Response. Thank you for this interesting suggestion. In response, we performed the analysis for the neoadjuvant therapy cohort, because sequential samples were available. The data is presented in new Extended Data Fig 2b,c and shows that the on-treatment biopsies do not add to the performance of the ROC of the logistic regression for response prediction.

Reviewer #3 (Remarks to the Author):

General Comment. *Valpione and colleagues investigate the TCR repertoire in melanoma biopsies from newly diagnosed patients and attempt to find correlates based on the hypothesis that the TCR repertoire of TIL could determine the disease course (clinical outcome). 16 patients recruited from 3 centers (Manchester, Padova, Meldola) received initial systemic therapy with anti-PD1 agents. The authors state that there were no differences in TIL numbers, TIL clones/clonotypes, TCR diversity, or TCR clonality (Extended data Fig 1). In addition, no correlation was found between OS and serum LDH, TIL numbers,*

TIL clones, TCR diversity, #SNVs, or PD-L1 expression levels (Extended data table 1). They did, however, find an association between OS and TCR clonality (Figure 1). Validation was assessed using datasets from Riaz (2017) and Yusko (2019) with melanoma patients (n=106) treated with either anti-PD1 or the combination anti-CTLA4/anti-PD1. The investigators then examined TCGA data base and found that TCR diversity (not clonality) as a prognostic marker correlated with OS for several cancers including BRCA, LUAD, TGCT, KIRK, THYM (Fig 2). The authors conclude: “high TIL/Tc clonality identifies which patients will mount an effective anti-PD1-induced immune response” and that this observation has important clinical implications.

This is an interesting observation; however, it is based on modest data with no mechanistic insight.

Response. We are delighted that the reviewer finds our observations interesting, we now comment on the limitations of our study in the Discussion (page 7, line 161)

Several major concerns are listed below.

Comment 1. *The authors studied 16 patients with metastatic melanoma. Please describe the clinical, pathologic, radiographic, and demographic features of these individuals. At first glance, the sample size appears exceedingly small and if this association (OS and high TCR clonality) is true, it should be broadened to be truly representative of newly diagnosed patients to convince one that others can build upon these results. The limited information in the data summary is inadequate.*

Response. We thank Reviewer #3 for the suggestion. Unfortunately, due to not-uniform methods for classification and missing response data, we could not analyse the relationship between TIL/Tc clonality and response for the cohorts originally presented in the first submission. Nevertheless, in response to the comment, we have added more detailed description of the patient clinical characteristics in new Table 1. Additionally, we now also include analysis of another three independent cohorts of melanoma melanoma patients treated with anti-PD1 immunotherapy in both the metastatic (Tumeh *et al.*, 2014, Roh *et al.*, 2017) and neoadjuvant setting (Amaria *et al.*, 2019). We present this new analysis in new Figure 1e-j and note that, despite the overall survival data for these cohorts not being available, this new data confirms that that high TIL/Tc clonality in the pre-treatment biopsies correlated with benefit from therapy.

Comment 2. *Data shown in Extended Data Figure 1 panel B reports the “total unique TCR sequences” for 6 data points from extranodal sites and 10 data points from nodal sites. The median number of unique TCR sequences in each group is 10 which is dramatically lower than expected based on the literature (and our own experience). For example, most tumor biopsies contain thousands or tens of thousands of unique clonotypes based on TCR VB CDR3 sequences obtained from melanoma patients (Pasetto CIR 2016; Riaz Cell 2017) or other histologies (Wu Nature 2020) using Adaptive Biotechnologies method of bulk sequencing gDNA or 10x Genomics sc mRNA platform. The low number of unique TCR clonotypes for each patient probably reflects a sampling bias or alternatively, a sequencing artifact due to their methodology. Finally, the authors should be more precise in their definition of a TCR clonotype and the primary sequencing data for all 16 subjects should be provided.*

Response. We thank the reviewer for these observations and agree that TCR sequence analysis from bulk RNA-Seq could lead to lower yield compared to other platforms. In response to the comment we have added an explorative study of TCR repertoire in the samples analysed through RNA-Seq and targeted sequencing and present this new data in new Extended Data Fig 3 and 4. The results show that although the number of unique clonotypes differed between the two methodologies, the overall calculated clonality did not. We are reassured by the consistency of our observations regarding the predictive value of TIL/Tc clonality as captured by multiple platforms, as now presented in Results (page 5 line 91), and also comment on this in the Discussion (page 7 line 146).

Comment 3. *The rationale to include TCGA data (Figure 2) from other cancer types is puzzling and some would argue it is extraneous to the observation made in figure 1. Namely, TCR clonality at baseline is predictive for clinical benefit after anti-PD1 therapy as first-line therapy. It would appear logical to investigate non-melanoma patients with newly diagnosed cancer administered anti-PD1 (either on study or as standard of care) to provide confirmatory evidence. The most pressing question for investigators in the field concerns the specificity of the high frequency TCRs evident in some patients and this was not addressed in the current study.*

Response. We thank the reviewer for this suggestion. Unfortunately, we do not have access to a cohort of newly diagnosed non-melanoma cancers treated with first-line anti-

PD1 immunotherapy and to the best of our knowledge, there are no published cohorts that we could use for this purpose. Moreover, although compelling, the study of the specificity of single TCR sequences requires *in-vitro* assays for each individual TCR, technically challenging analysis that falls beyond the scope of the current study. Nevertheless, in response we have now included new analyses showing that the TIL/Tc sequences were broadly private to individual samples and thus unlikely to recognise a shared antigen, and we present this data in new Extended Data Fig 5.

Minor Points

Point 1. *A point worth mentioning relates to an assumption implicit in the authors conclusion that “an effective anti-PD1-induced immune response” somehow equates with improved OS. The data in Figure 1 concerns OS as a clinical endpoint. The authors show no data related to the quality, composition or magnitude of the T cell response after anti-PD1 treatment.*

Response. We thank Reviewer #3 for this observation. Although our study focused on pre-treatment TIL/Tc repertoire and the analysis of the composition and the magnitude of T cell responses after anti-PD1 was beyond the scope of the current study, in response to the comment we have now added three new cohorts that allowed us to study the radiological response to treatment. This new analysis shows that high pre-treatment TIL/Tc correlates with better response, and we present this new data in new Figures 1e-j and describe the Result in the text (page 4 line 80).

Point 2. *The authors appear to state that a prospective analysis of anti-PD1 naïve patients is no longer feasible since adjuvant administration is now standard of care. I would be cautious about this conclusion as many stage II (and a small percentage of stage I) patients will not develop nodal regional disease but rather proceed to stage IV metastatic disease that can be biopsied to assess TCR clonality prior to first line anti-PD-1.*

Response. We thank Reviewer #3 for this comment, and in response we have rephrased this section and also commented on the fact that adjuvant anti-PD1 clinical trials are currently ongoing in stage II melanoma (page 7 line 151).

Reviewers' Comments:

Reviewer #1:

Remarks to the Author:

In this revised version of the manuscript the authors have adequately addressed my previously raised concerns. I support the publication of this work.

Reviewer #2:

Remarks to the Author:

The authors have satisfactorily addressed this reviewer's concerns in their revised manuscript.

Reviewer #3:

Remarks to the Author:

I continue to have reservations about the use of RNA-Seq to infer TCR repertoire especially when targeted methods such as Adaptive Biotechnologies platform are readily available to reliably capture the near-complete BV repertoire (see P. Barennes et. al. Sept 2020 Nature Biotech). In addition, the lack of mechanistic insight is a weakness.

Comment 1 requested additional details regarding the study population of 16 patients. In response, the authors provided Table 1 which provides limited clinical details. In addition, Figure 1e-j is added to the revised manuscript which contains data from 3 published melanoma cohorts that reported ORR yet lacks OS endpoint.

Comment 2 addressed the small number (median 10.5) of clonotypes inferred using ImReP (developed for Ig clonotypes) and the authors responded by adding new Figures 3 and 4 which compares RNA-seq (n=16 samples) to targeted BV sequencing (Adaptive Bio platform) of samples from ref 3,4,6,7. A better comparison would have been to compare RNA-seq (ImReP) and a targeted BV assessment using the same 16 samples.

Comment 3 addressed the claim that TCR diversity is prognostic for OS in several other solid tumors. It is unclear how this observation made using TCGA samples is relevant to the melanoma work.

We are grateful to the reviewers for the time spent on our manuscript. Their comments from the first review were both insightful and helpful, and in responding, the quality of the manuscript improved. We are delighted that Reviewers #1 and #2 have recommended publication of the revised manuscript. We are however somewhat surprised by the new requests from Reviewer #3, which arose from the second round of review. Specifically, addressing these new requests would not affect the main ideas or data quality in our manuscript and below, we outline our point-by-point responses to the specific requests. We ask your forbearance in this matter, because if we did address the new requests, it would not strengthen the manuscript any further, but it would cause us unnecessary delay and significant expense.

REVIEWER COMMENTS

Reviewer #1 (Remarks to the Author):

Comment: *In this revised version of the manuscript the authors have adequately addressed my previously raised concerns. I support the publication of this work.*

Response: We thank Reviewer #1 for suggesting the improvements in the first round of review, and we are delighted that the reviewer is satisfied with our responses and has recommended publication of the manuscript.

Reviewer #2 (Remarks to the Author):

Comment: *The authors have satisfactorily addressed this reviewer's concerns in their revised manuscript.*

Response: Again, we are grateful to Reviewer #2 for the suggestions in the first round of review and are delighted that this reviewer too is satisfied with our responses and has also recommended publication.

Reviewer #3 (Remarks to the Author):

Comment: *I continue to have reservations about the use of RNA-Seq to infer TCR repertoire especially when targeted methods such as Adaptive Biotechnologies platform are readily available to reliably capture the near-complete BV repertoire (see P. Barennes et. al. Sept 2020 Nature Biotech). In addition, the lack of mechanistic insight is a weakness.*

Response: We thank Reviewer #3 for their continued consideration of our manuscript. However, we wish to emphasise that the point of our study is to show that the clonality of tumour infiltrating T cells is predictive of benefit from anti-PD1 therapy. We do not seek to endorse RNA-Seq-based algorithms as an approach to reconstruct TCR sequences of tumour infiltrating T cells, as this has already been done elsewhere by those who developed the pipeline. Moreover, an in-depth mechanistic analysis is beyond the scope of our manuscript, the purpose of which is to explore how personalised immunotherapy could be delivered.

More pertinent, please note that 170 of of 186 samples we present were analysed on the Adaptive Biotechnology platform that Reviewer #3 favours. The results consistently indicate that tumour infiltrating T cell clonality predicts outcome to immune-checkpoint blockade. Specifically, using two independent cohorts, we show that high tumour infiltrating T cell clonality is predictive for better OS in 122 patients, and moreover that, in 3 additional independent cohorts it is predictive of better radiological response in 61 patients. We wish also to highlight that in the revised manuscript, and in response to the requests of the other reviewers, we provide a comparison of the T cell repertoire metrics inferred using the two TCR analysis platforms and that the distribution of the predictive biomarker – the tumour

infiltrating T cell clonality – is not skewed in the two approaches (Extended Data Fig 3a). We fail therefore to understand how repeating the Adaptive Biotechnologies analysis on 16 of the 186 samples could affect the key concepts of our manuscript or, indeed, increase the quality of the data.

Comment: *Comment 1 requested additional details regarding the study population of 16 patients. In response, the authors provided Table 1 which provides limited clinical details. In addition, Figure 1e-j is added to the revised manuscript which contains data from 3 published melanoma cohorts that reported ORR yet lacks OS endpoint.*

Response: Please note that the clinical details provided in Table 1 are presented in accordance with commonly reported descriptive variables for melanoma patient clinical data. We are happy to add further variables if we receive specific and justified requests, but they would need to be within the limits of patient anonymisation requirements (GDPR).

Comment: *Comment 2 addressed the small number (median 10.5) of clonotypes inferred using ImReP (developed for Ig clonotypes) and the authors responded by adding new Figures 3 and 4 which compares RNA-seq (n=16 samples) to targeted BV sequencing (Adaptive Bio platform) of samples from ref 3,4,6,7. A better comparison would have been to compare RNA-seq (ImReP) and a targeted BV assessment using the same 16 samples.*

Response: This is a new request for us to re-analyse our original 16 test samples with the Adaptive Biotechnology platform. However, we cannot see a compelling justification to do so, because 4 out of the 5 cohorts, representing 170 of the 186 patient samples we present were analysed using the Adaptive Biotechnology platform that Reviewer #3 favours. Note also that the results across the 5 cohorts consistently indicate that high T cell clonality in pre-treatment biopsies predicts better outcomes to immune-checkpoint blockade. In particular, the OS findings on the 16 patients analysed by RNA-seq have been fully validated by the external cohort of 106 patients analysed by the Adaptive Biotechnology platform (Figure 1c,d).

We understand of course that with evolving technologies, more sensitive and accurate solutions will eventually replace existing state-of-the-art platforms. However, an Adaptive Biotechnology re-analysis of only 16 of our 186 samples (less than 10%) will not affect the main concept of the study or the data quality of our manuscript. It is therefore unclear to us what the additional analysis would add, apart from causing us unnecessary very high costs and time delays in this highly competitive field. Moreover, performing this analysis is made all the more complicated by the current COVID lockdown situation.

Comment: *Comment 3 addressed the claim that TCR diversity is prognostic for OS in several other solid tumors. It is unclear how this observation made using TCGA samples is relevant to the melanoma work.*

Response: We wish to emphasise that the purpose of our analysis of TCGA cohorts is to support the idea that tumour infiltrating T cell diversity is prognostic in tumours not treated with immunotherapy, thus endorsing the idea that tumour infiltrating T cell clonality is a predictive biomarker specific for immune-checkpoint blockade. This analysis demonstrates that our observation has broad ramifications, not only in the clinical implications for treatment decisions for immune-checkpoint blockade in the adjuvant setting, but also because it is confirmed across multiple cancer cohorts and thus supports the hypothesis that the mechanisms of immune-surveillance that determine cancer prognosis are different from the determinants of response to immune-checkpoint blockade. The broader ramifications of our study will therefore be of interest to the wide readership of *Nature Communications*, and

particularly to those outside the melanoma field. We prefer therefore to retain this analysis in the manuscript.

Reviewers' Comments:

Reviewer #4:

Remarks to the Author:

The manuscript addresses a question which is very relevant in the field, indeed the availability of both TCR reconstruction methods and of single cell TCR sequencing approaches is opening up new possibility for the interpretation of data on clonal expansion at tumor sites.

Moreover, the need for a better understanding of the molecular mechanisms underlying immune checkpoint inhibitor treatments is paralleled by the need of best prognostic markers to select the patients that will benefit more from the therapy.

Data and methodologies are carefully presented and the Authors are cautious in the interpretation of their findings. Of course, the study is highly correlative and it opens many questions (e.g. the nature of the clonally expanded subsets) but I reckon that the key point here is the proposal of TIL/Tc repertoire as a biomarker for improved patient selection. The Authors are not trying to provide an explanation for the interesting concept that clonal expansion is related to successful therapeutic outcome, which would, in my opinion, go beyond the scope of the present work.